# Exosomal miR-224 contributes to hemolymph microbiota homeostasis during bacterial infection in crustacean

Yi Gong[1,2,3], Xiaoyuan Wei[1,3], Wanwei Sun[1,3], Xin Ren[1,3], Jiao Chen[1,3], Jude Juventus Aweya[1,2,3], Hongyu Ma[1,2,3], Kok-Gan Chan[3,4], Yueling Zhang[1,2,3]*, Shengkang Li[1,2,3]*

1 Guangdong Provincial Key Laboratory of Marine Biology, Shantou University, Shantou, China, 2 Southern Marine Science and Engineering Guangdong Laboratory, Guangzhou, China, 3 Institute of Marine Sciences, Shantou University, Shantou, China, 4 Division of Genetics and Molecular Biology, Institute of Biological Science, Faculty of Science, University of Malaya, Kuala Lumpur, Malaysia

* zhangyl@stu.edu.cn (YZ); lisk@stu.edu.cn (SL)

**Data Availability Statement:** Exosomal miRNAs were identified through miRNA sequencing based on Illumina platform by a commercial company

## Abstract

It is well known that exosomes could serve as anti-microbial immune factors in animals. However, despite growing evidences have shown that the homeostasis of the hemolymph microbiota was vital for immune regulation in crustaceans, the relationship between exosomes and hemolymph microbiota homeostasis during pathogenic bacteria infection has not been addressed. Here, we reported that exosomes released from *Vibrio parahaemolyticus*-infected mud crabs (*Scylla paramamosain*) could help to maintain the homeostasis of hemolymph microbiota and have a protective effect on the mortality of the host during the infection process. We further confirmed that miR-224 was densely packaged in these exosomes, resulting in the suppression of HSP70 and disruption of the HSP70-TRAF6 complex, then the released TRAF6 further interacted with Ecsit to regulate the production of mitochondrial ROS (mROS) and the expression of Anti-lipopolysaccharide factors (ALFs) in recipient hemocytes, which eventually affected hemolymph microbiota homeostasis in response to the pathogenic bacteria infection in mud crab. To the best of our knowledge, this is the first document that reports the role of exosome in the hemolymph microbiota homeostasis modulation during pathogen infection, which reveals the crosstalk between exosomal miRNAs and innate immune response in crustaceans.

## Author summary

Exosomes are small membrane vesicles of endocytic origin which are widely involved in the regulation of a variety of pathological processes in mammals. Yet, although the anti-bacterial function of exosomes has been discovered for many years, the relationship between exosomes and hemolymph microbiota homeostasis remains unknown. In the present study, we identified the miRNAs packaged by exosomes that were possibly involved in *Vibrio parahaemolyticus* infection by modulating hemolymph microbiota homeostasis in crustacean mud crab *Scylla paramamosain*. Moreover, it was found that

(Biomarker Technologies, Beijing, China), the original sequencing data were uploaded to NCBI BioProject database with accession number PRJNA600674. Hemolymph bacteria were sequenced on Illumina Nova platform by a commercial company (Novogene, Beijing, China), the data were uploaded to NCBI BioProject database (accession number PRJNA669103).

**Funding:** This study was financially supported by 2020 Li Ka Shing Foundation Cross-Disciplinary Research Grant (2020LKSFG01E) (https://lksf.org/) (Received by YLZ), Key Special Project for Introduced Talents Team of Southern Marine Science and Engineering Guangdong Laboratory (Guangzhou) (GML2019ZD0606) (http://www.gmlab.ac.cn/) (Received by SKL), National Natural Science Foundation of China (31802341, 42076125, 41876152) (http://www.nsfc.gov.cn/) (31802341 is received by YG, 42076125 and 41876152 are received by SKL), Natural Science Foundation of Guangdong Province, China (2018A030307044) (http://gdstc.gd.gov.cn) (Received by YG) and Guangdong Provincial Special Fund for Modern Agriculture Industry Technology Innovation Teams (2019KJ141) (http://dara.gd.gov.cn/) (Received by SKL). The funders had no role in study design, data collection and analysis, decision to publish, or preparation of the manuscript.

**Competing interests:** The authors have declared that no competing interests exist.

miR-224 was densely packaged in exosomes after *Vibrio parahaemolyticus* challenge, resulting in the suppression of HSP70 and disruption of the HSP70-TRAF6 complex in recipient hemocytes, then the released TRAF6 was further interacted with Ecsit to regulate ROS and ALFs levels, which eventually affected hemolymph microbiota homeostasis to cope with pathogenic bacteria infection. Our finding is the first to reveal the relationship between exosomes and hemolymph microbiota homeostasis in animals, which shows a novel molecular mechanism of invertebrate resistance to pathogenic microbial infection.

## Introduction

Exosomes are nanosized (30–150 nm in diameter) extracellular vesicles formed in multivesicular bodies and released into the extracellular environment under physiological and pathological conditions [1,2]. Specific proteins highly enriched in exosomes such as TSG101, CD63, CD81 and flotillin 1, usually serve as indicators for the identification of exosomes [3]. Exosomes can be secreted by various donor cells and transferred to target cells by fusing with cytomembranes, which serve as mediators during intercellular communications via transporting bio-cargoes, such as nucleic acids, proteins and lipids [4,5]. Given their role as a form of intercellular vesicular transport, numerous studies have pointed out the importance of exosomes during pathogen infection and immune response [6,7]. It is believed that pathogen-infected cells are capable of secreting exosomes that contain pathogens or host genetic elements to neighboring cells to help modulate host immune response, which has huge impact on the fate of the infection process [8,9]. However, little is known about how exosomes regulate host immune response and impact on pathogen infection, especially in crustaceans.

MicroRNAs (miRNAs), a class of small non-coding RNAs with 18–25 nucleotides in length, can interact with the complementary sequences on the 3' untranslated region (3'UTR) of target mRNA to either arrest translation or degrade the mRNA of the target genes [10,11]. Apart from their endogenous functions, miRNAs can be packaged into exosomes to modulate the expression of specific target genes in recipient cells [12,13]. Furthermore, recent studies have revealed that loading of miRNAs into exosomes is a selective process and can reflect the dysregulated miRNA composition in donor cells [14]. It has been demonstrated that alteration of exosomal miRNA composition has great influence on the biological activities of exosomes that have been taken-up during pathogens invasion [15,16]. It is thought that exosome-mediated intercellular transfer of miRNAs can regulate pathogens spread and immune defense in recipient cells, which suggest that exosomal miRNAs could play potential role as novel tools for intercellular communication.

Crustaceans have an open circulatory system, where hemocytes, oxygen, hormones and nutrients circulate together in the hemolymph [17]. Symbiotic microorganisms are indispensable inhabitants in the host [18], with growing evidence showing the presence of diverse microorganisms in the hemolymph of aquatic invertebrates including shrimp [19], scallop [20] and crab [21]. Generally, the proliferation of microbiota in the nutrient rich hemolymph environment is tightly controlled by host immune factors such agglutination, phagocytosis, production of antimicrobial peptides and reactive oxygen species (ROS) [19,22]. For instance, it has been shown that a shrimp C-type lectin, *Mj*HeCL, maintains hemolymph microbiota homeostasis by modulating the expression of antimicrobial peptides [23]. Hemolymph symbiotic microbiota in hemolymph is believed to be engaged in multiple functions in the host, including competing with invading pathogens or stimulating the host to mount an immune response during pathogens infection [24,25]. Unfortunately, uncontrolled proliferation of

hemolymph microbiota could result in host diseases such as "Milky Disease" or "Early Mortality" [26,27], which highlights the importance of hemolymph microbiota in host immune system and disease prevention.

The open circulatory system of crustaceans makes it an ideal carrier for exosomes to perform immune-related functions. However, the role of exosomes in maintaining hemolymph microbiota homeostasis remains unclear. In the light of this, the current study explored the relationship between exosomes and hemolymph microbiota in mud crab. Exosomes released from *V. parahaemolyticus*-infected mud crabs could reduce crab mortality due to *V. parahaemolyticus* infection by maintaining the homeostasis of hemolymph microbiota. Moreover, miR-224 was found to accumulate in exosomes after *V. parahaemolyticus* infection, which resulted in suppression of heat shock protein 70 (HSP70) and disruption of the HSP70-TNF receptor associated factor 6 (TRAF6) complex. The released TRAF6 then interacted with evolutionarily conserved signaling intermediate in Toll pathways (Ecsit) to regulate mitochondrial ROS (mROS) production and the expression of anti-lipopolysaccharide factors (ALFs) in the recipient hemocytes, which eventually affected hemolymph microbiota homeostasis in response to the infection.

## Results

### The involvement of exosomes in anti-bacterial response in mud crab

To explore the involvement of exosomes from mud crab in bacterial infection, exosomes were isolated from the hemolymph of *V. parahaemolyticus*-challenged mud crabs (i.e., exosome-Vp) and Phosphate buffer saline (PBS)-injected control crabs (i.e., exosome-PBS). After that, a series of tests required [28] were performed to evaluate the quality of the isolated exosomes. The typical cup-shaped structures of isolated exosomes were observed under an electron microscope (Fig 1A) and their sizes were measured by Nanosight particle tracking analysis (Fig 1B). The isolated particles were further ascertained as exosomes by determining the exosomal protein markers Flotillin-1, TSG101 and the cytoplasmic marker (Negative control) Calnexin using Western blot analysis (Fig 1C). These results indicate successful isolation of exosomes from mud crabs challenged with *V. parahaemolyticus* and PBS, and there is no major difference between them.

Next, the ability of the isolated exosomes to be internalized by mud crab hemocytes was analyzed by labeling the isolated exosomes with DiO (green) before being injected into mud crabs. When hemocytes from the injected crabs were collected and labeled with DiI (red) before being examined with a confocal laser scanning microscope, the results showed that the isolated exosomes could be internalized in hemocytes (Fig 1D). Moreover, through flow cytometry analysis, we found that exosome-Vp and exosome-PBS possess similar binding activity to the recipient hemocytes (S1 Fig). To explore the involvement of exosomes in mud crab during pathogenic bacteria infection, the isolated exosomes (exosome-Vp and exosome-PBS) were mixed with *V. parahaemolyticus* before being injected into mud crabs to determine the mortality rate, Wild Type (WT) or *V. parahaemolyticus*-treated mud crabs serve as control groups. As shown in Fig 1E, there was significant reduction in the mortality rate of mud crabs injected with exosome-Vp mixed with *V. parahaemolyticus* compared with mud crabs injected with exosome-PBS mixed with *V. parahaemolyticus*, which suggested that exosomes isolated from *V. parahaemolyticus*-challenged mud crabs have a protective effect on pathogenic bacteria infection. Moreover, when the relative abundance of hemolymph bacteria in mud crabs was determined, the results revealed that exosome-Vp was able to inhibit the rapid increase in hemolymph bacteria during infection (Fig 1F). These data of bacteria abundance and mortality rate showed that exosome-Vp possesses a protective effect on the mortality and proliferation

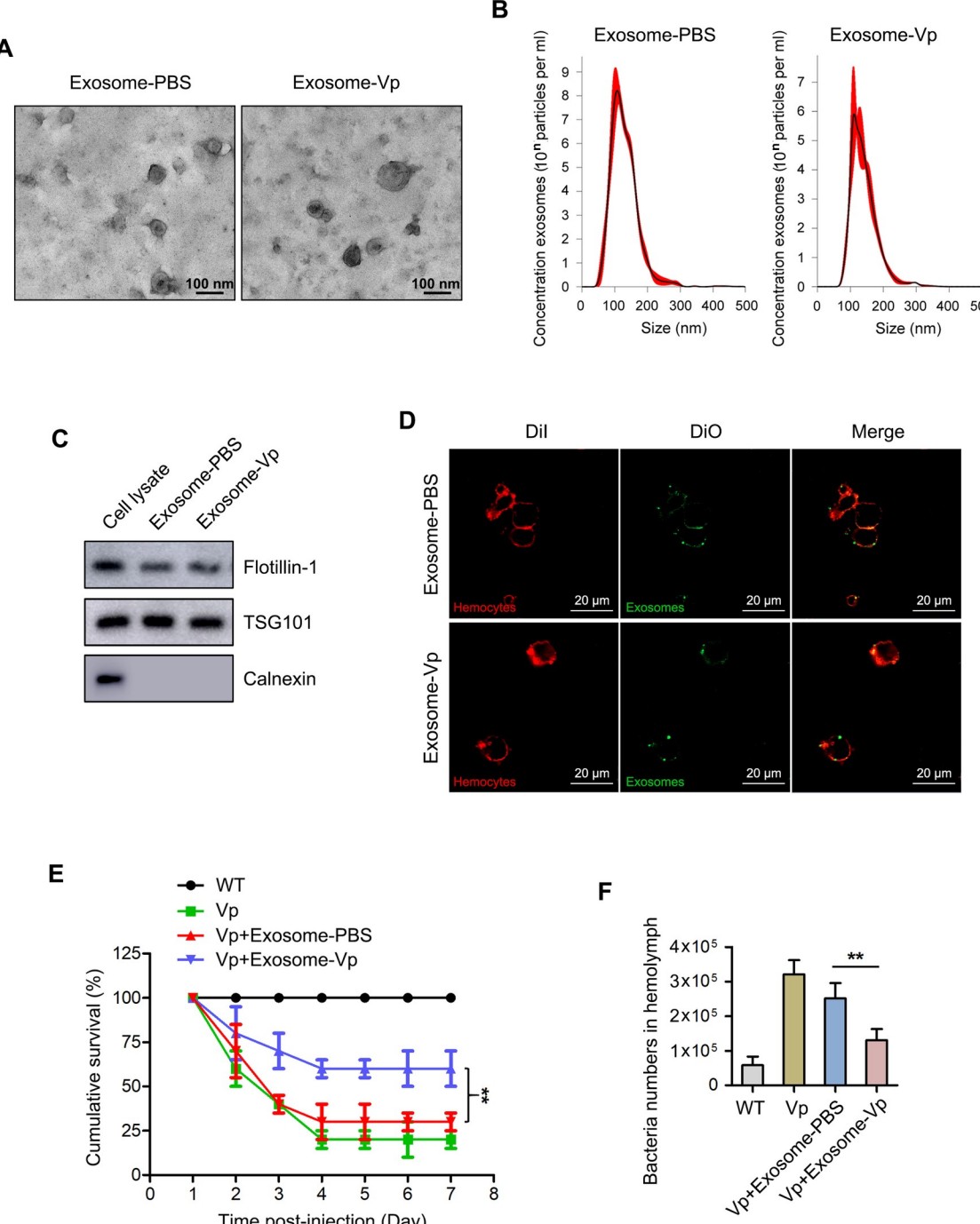

**Fig 1. Exosomes secreted from *Vibrio parahaemolyticus*-infected mud crab participate in anti-bacterial regulation. (A-B)** Exosomes isolated from mud crabs injected with PBS and *V. parahaemolyticus* were detected by electron microscopy **(A)** and Nanosight particle tracking analysis **(B)**. Scale bar, 200 nm. **(C)** Western blot analysis of exosomal protein markers (Flotillin-1 and TSG101) and cytoplasmic marker (Negative control) Calnexin in cell lysate and exosomes. **(D)** The delivery of exosomes to mud crab hemocytes. The indicated exosomes (Dio-labeled, green) were injected into mud crabs for 6 h, after which hemocytes (DiI-labeled, red) were isolated and analyzed by confocal microscopy. Scale bar, 20 μm. **(E)** Effects of exosomes on mud crab mortality. The specific treatments are shown on the top and the mortality was examined daily. **(F)** Effects of exosomes on bacteria number in mud crab hemolymph. Hemolymph bacteria number for the different treatments were counted using a fluorescence microscope at 100× magnification. (Vp means *V. parahaemolyticus*, exosome-Vp or exosome-PBS means exosomes isolated from the hemolymph of crabs challenged with *V. parahaemolyticus* or PBS). Significant statistical differences between treatments are indicated with asterisks (**, $p < 0.01$).

of bacteria in the hemolymph, which may suggest a role in microbiota homeostasis and immune response. Taken together, these results suggest that exosomes secreted by *V. parahaemolyticus*-challenged mud crabs play a role in anti-bacterial response in mud crabs, probably by helping to maintain homeostasis of hemolymph microbiota.

### Exosomes modulate hemolymph microbiota homeostasis

To ascertain the regulatory function of exosomes in modulating the mud crab hemolymph microbiota homeostasis, we determined the expression of antimicrobial peptides (AMPs) and ROS level, which are essential in regulating hemolymph microbiota homeostasis [19,22]. The results revealed that exosome-Vp treatment could significantly increase ROS levels in mud crabs during pathogenic bacteria infection compared with the exosome-PBS (Fig 2A and 2B). Similarly, transcript levels of antimicrobial peptides (ALF1, ALF4 and ALF5) were significantly increased in mud crabs treated with exosome-Vp (Fig 2C). Next, the bacteria species and composition of hemolymph microbiota were analyzed using 16S rDNA sequencing. As shown in Fig 2D, the diversity of hemolymph microbiota in mud crabs decreased significantly during *V. parahaemolyticus* infection. However, hemolymph microbiota diversity was maintained during the infection following treatment of mud crabs with exosome-Vp as compared with exosome-PBS. When the composition of hemolymph microbiota was analyzed at the phylum level, the proportion of *Proteobacteria*, *Tenericutes* and *Firmicutes* increased during *V. parahaemolyticus* infection, while the proportion of *Acidobacteria*, *Actinobacteria* and *Chloroflexi* decreased. On the contrary, when mud crabs were treated with exosome-Vp, microbiota homeostasis was maintained in mud crabs during the infection (Fig 2E). A similar trend was observed when the top 35 genera of hemolymph microbiota was analyzed (Fig 2F). Meanwhile, it was found that some bacteria highly presented during *V. parahaemolyticus* (*Lactobacillus*, *Moheibacter*, etc) were decreased in the presence of both exosome-Vp and exosome-PBS, which might be mediated by the common bio-molecules packaged in them (Fig 2F). All these results suggest that during pathogenic bacteria infections in mud crabs, exosomes could maintain the homeostasis of hemolymph microbiota probably by regulating the levels of mROS and ALFs.

### Functional miRNA screening in exosomes

To determine the functional exosomal miRNAs that are crucial in modulating hemolymph microbiota homeostasis, miRNA sequencing was carried out on exosome-Vp and exosome-PBS. Among the differentially expressed exosomal miRNAs, the top 6 miRNAs (Fig 3A) which include miR-291, miR-343, miR-224, miR-189, miR-60 and miR-156 were selected to investigate their role in *V. parahaemolyticus* infection in mud crabs. To screen for the potential functional miRNAs in exosome-Vp, ALF1 was used as the indicator. The miRNA mimics and anti-miRNA oligonucleotides (AMOs) of these miRNAs were synthesized and co-injected with *V. parahaemolyticus* into mud crabs followed by qPCR analysis of ALF1 expression. The results revealed that injection of mud crabs with miR-224 mimics increased the expression of ALF1, while injection with AMO-miR-224 decreased the ALF1 expression (Fig 3B and 3C).

To ascertain whether exosome-Vp was involved in regulating hemolymph microbiota homeostasis via exosomal miR-224, the relative expression level of miR-224 was determined in exosome-Vp and exosome-PBS injected mud crabs. The results revealed significant upregulation in the expression of miR-224 in the exosome-Vp injected mud crabs compared with exosome-PBS (Fig 3D), indicating that exosome-Vp treatment could led to miR-224 accumulation in the recipient cells. Next, the involvement of miR-224 in the exosome-mediated regulatory process was examined by co-injecting *V. parahaemolyticus* with exosome-PBS,

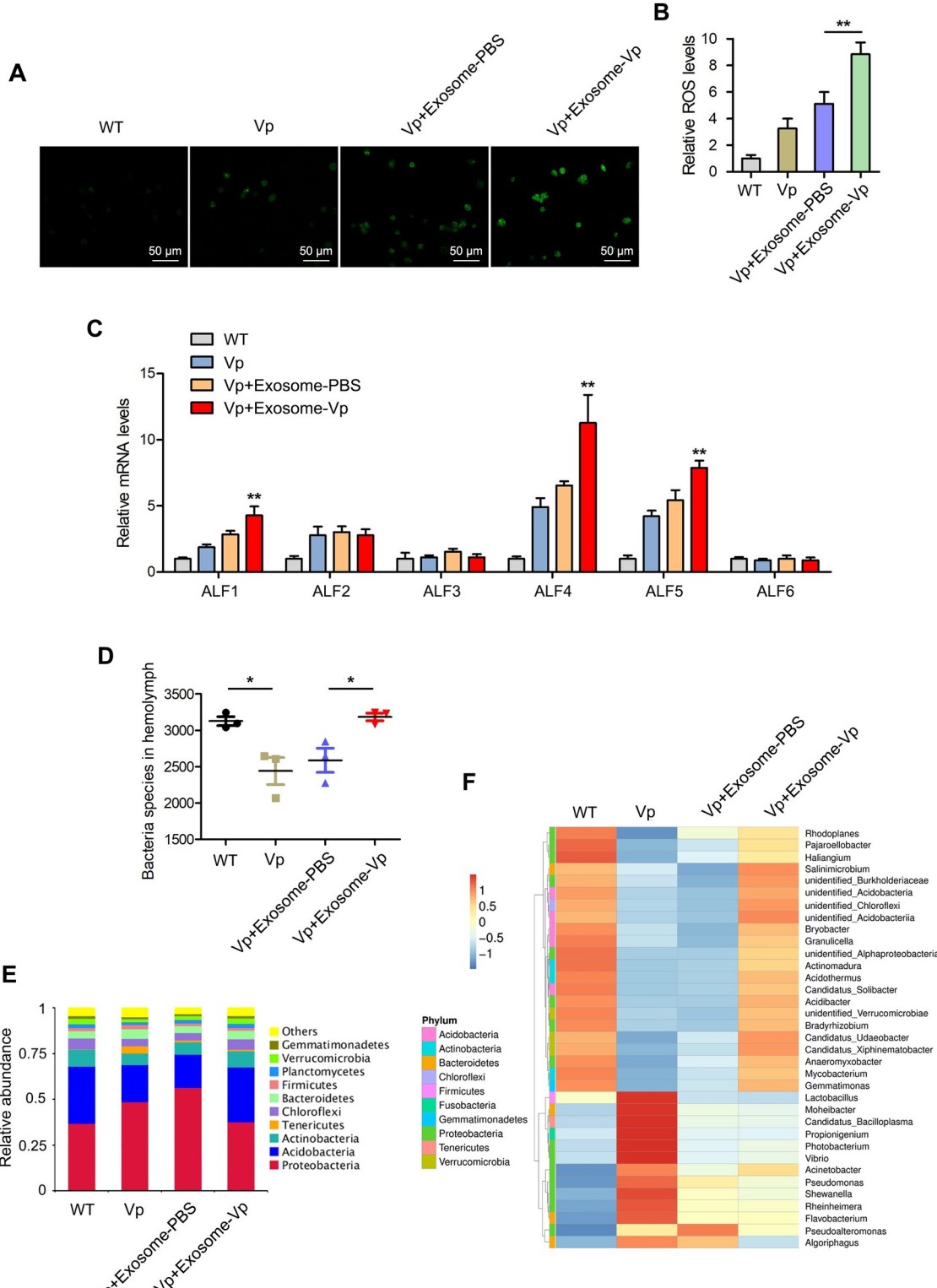

**Fig 2. Exosomes regulate hemolymph microbiota homeostasis through activation of ROS and ALFs. (A-B)** The effects of the indicated exosomes on ROS production during *V. parahaemolyticus* infection in mud crabs. The level of ROS was measured by fluorescence microscopy. Scale bar, 50 μm **(A)** and microplate reader **(B)**. **(C)** The effect of exosomes on the mRNA levels of ALF1-ALF6, and β-actin was used as internal reference, each treatment contains 5 crabs and three independent experiments were performed. **(D)** The effects of the indicated exosomes on hemolymph microbiota diversity. Mud crabs were co-injected with

exosomes and *V. parahaemolyticus* for 48 h, after which hemolymph was collected and subjected to 16S rDNA sequencing. **(E-F)** The effects of the indicated exosomes on the composition of hemolymph microbiota at phylum (Top 10) **(E)** and genera (Top 35) **(F)** levels. Data represent mean ± s.d. of triplicate assays (*, $p<0.05$; **, $p<0.01$).

exosome-Vp or exosome-Vp and AMO-miR-224 into the mud crabs. The level of ROS in the exosome-Vp and AMO-miR-224 co-injected mud crabs was significantly lower compared with the other mud crabs (Fig 3E and 3F). A similar trend was observed in the expression levels of ALF1, ALF4 and ALF5 for these mud crab samples (Fig 3G). In addition, 16S rDNA sequencing analysis revealed a disruption in the exosome-Vp-mediated hemolymph microbiota homeostasis upon miR-224 silencing (Fig 3H and 3I). These results suggest that *V. parahaemolyticus*-derived exosomes could help to maintain hemolymph microbiota homeostasis of mud crab by miR-224, a miRNA densely packaged in exosomes after *V. parahaemolyticus* challenge.

## Interactions between miR-224 and its target gene

To explore the pathways mediated by miR-224 in mud crab, the genes targeted by miR-224 were predicted by Targetscan and miRanda software. The prediction revealed that HSP70 was the only target gene predicted by both software (Fig 4A), moreover, the sequence alignment results indicated that the seed sequence of miR-224 could be completely complementary with the 3'UTR of HSP70 mRNA (Fig 4A). Therefore, HSP70 was preferred as the potential target gene regulated by miR-224. To ascertain this prediction, synthetic miR-224 and EGFP-HSP70-3'UTR or the mutant EGFP-ΔHSP70-3'UTR were co-transfected into *Drosophila* S2 cells [29], EGFP-HSP70-3'UTR only was used as control group (Fig 4B). When the EGFP fluorescence activity of these transfectants was observed under a fluorescence microscopy and a microplate reader, a significant decrease in fluorescence intensity was observed in cells co-transfected with EGFP-HSP70-3'UTR compared with control (Fig 4C and 4D), which indicates that miR-224 potentially interacts with HSP70 to modulate its expression.

To investigate the interaction between miR-224 and HSP70 in mud crabs, miR-224 was silenced or overexpressed followed by HSP70 detection. The results revealed significant increase in both mRNA and protein levels of HSP70 after AMO-miR-224 treatment (Fig 4E and 4F). On the contrary, the mRNA and protein levels of HSP70 decreased upon miR-224 overexpression (Fig 4G and 4H). Furthermore, fluorescence *in situ* hybridization (FISH) analysis was carried out to determine the subcellular location of miR-224 and HSP70 in mud crabs hemocytes. When miR-224 and HSP70 mRNA were labeled with fluorescent probes before being observed under a confocal microscopy, miR-224 and HSP70 mRNA were found to colocalize in hemocytes of the mud crabs (Fig 4I), and Tubulin mRNA were used as a negative control (Fig 4I). All these results suggest that HSP70 is the direct target gene of miR-224 in the mud crabs.

## Effect of HSP70 on the modulation of hemolymph microbiota homeostasis

To ascertain whether HSP70 is involved in the modulation of miR-224-mediated hemolymph microbiota homeostasis, miR-224-depleted mud crabs were injected with HSP70-siRNA before being infected with *V. parahaemolyticus* and the levels of ALFs and ROS were detected in hemocytes. The results revealed significant increase in the expression of ALF1, ALF4 and ALF5 in the HSP70-siRNA treated group compared with controls (Fig 5A). Similar results were obtained for ROS levels (Fig 5B and 5C). Moreover, the expression of HSP70 was significantly decreased in exosome-Vp treated mud crabs as compared with control (Fig 5D), which indicates that HSP70 participates in exosome-mediated regulatory process. Besides, in

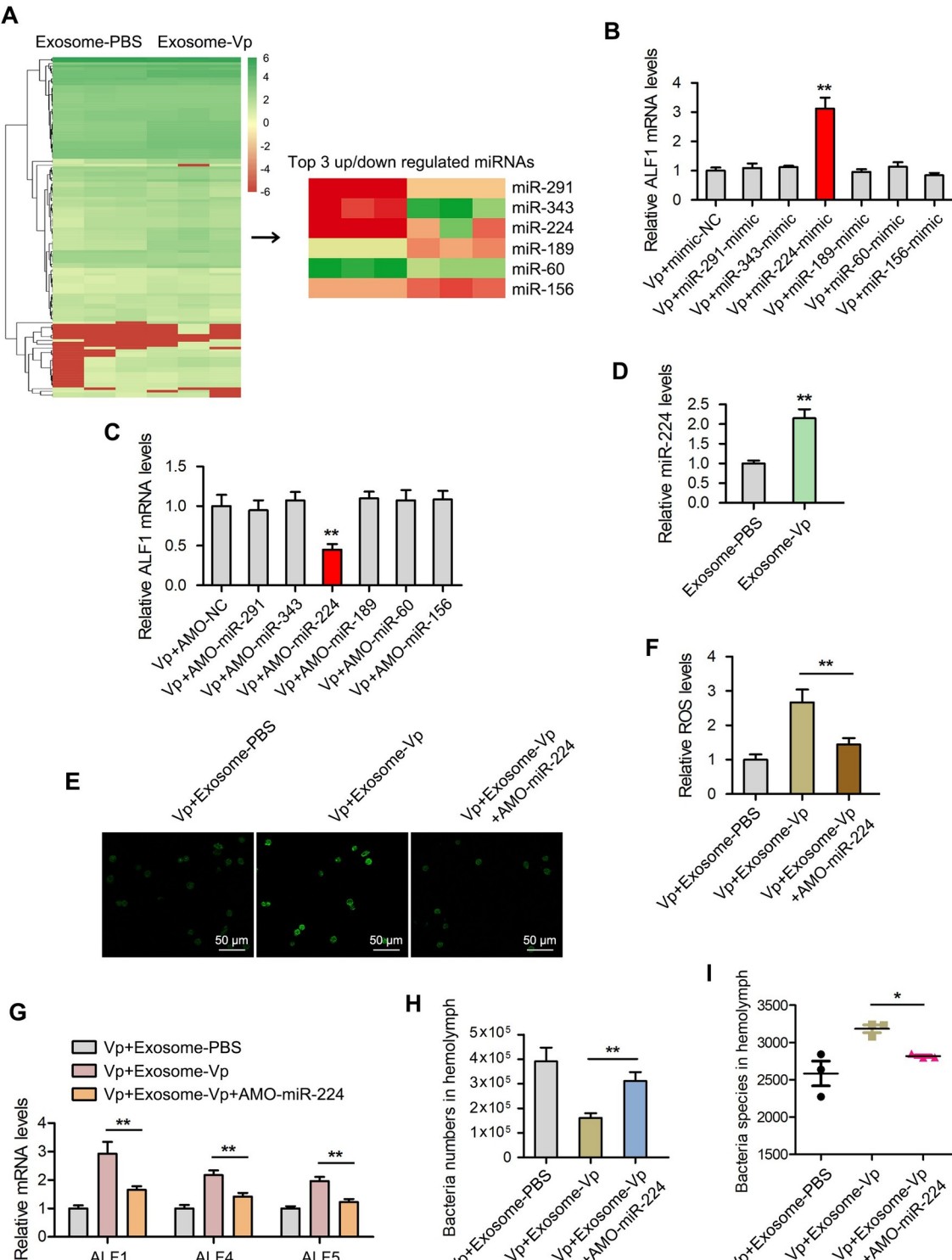

**Fig 3. Exosomal miR-224 modulates hemolymph microbiota homeostasis in mud crabs. (A)** miRNA sequencing analysis for exosome-V.p and exosome-PBS is presented as a heatmap. The top three up- and downregulated miRNAs in the indicated exosomes are listed. **(B-C)** The effects of the indicated miRNAs on ALF1 expression in mud crabs. Mimics **(B)** or AMOs **(C)** of the indicated miRNAs were co-injected with *V. parahaemolyticus* into mud crabs for 48 h, followed by the analysis of ALF1 expression using qPCR. Data were presented relative to the value of Vp+ mimic-NC or Vp+ AMO-NC group, which were treated as standard "1". mimic-NC and AMO-NC stand for disordered nucleic acids which serve as the negative controls. **(D)** The expression levels of miR-224 in mud crabs challenged with different exosomes, U6 was used as an internal reference. **(E-F)** The participation of miR-224 in exosome-

mediated ROS production. The indicated exosomes, AMO-miR-224 and *V. parahaemolyticus* were co-injected into mud crabs, followed by the detection of ROS using fluorescence microscopy, Scale bar, 50 μm **(E)** and microplate reader **(F)**. **(G)** The effect of miR-224 silencing on exosome-mediated ALFs regulation. **(H-I)** The involvement of miR-224 in exosome-mediated hemolymph microbiota homeostasis. Hemolymph was collected from mud crabs with the indicated treatments, following by determining the bacterial cell count **(H)** and species **(I)** analysis. The data of Vp+Exosome-PBS and Vp+Exosome-Vp groups were from Fig 2D. Each experiment was performed in triplicate and data are presented as mean ± s.d. (*, $p < 0.05$; **, $p < 0.01$).

HSP70-depleted mud crabs co-injected with *V. parahaemolyticus* and exosome-PBS, there were lower hemolymph bacteria numbers but higher hemolymph bacteria diversity (Fig 5E and 5F), which suggest that exosome-PBS could also maintain hemolymph microbiota homeostasis when the expression of HSP70 is suppressed. These results suggest that exosomal miR-224 contributes to hemolymph microbiota homeostasis by targeting HSP70 in mud crabs.

## Formation of the TRAF6-Ecsit complex in the exosomal regulatory pathway

Based on the observation that HSP70 was relevant in the modulation of hemolymph microbiota homeostasis, pull-down analysis was carried out followed by SDS-PAGE and Western blot analyses. The results showed that HSP70 could bind to TRAF6 (Fig 6A and 6B and S1 Table). The role of TRAF6 in exosomal miR-224-mediated regulatory process was investigated by silencing the expression of TRAF6 in mud crabs and the expression of ALFs detected by qPCR analysis. As shown in Fig 6C, the expression levels of ALF1, ALF4 and ALF5 were significantly decreased. Similarly, there was significant decrease in ROS level in hemocytes of TRAF6-silenced mud crabs compared with control (Fig 6D and 6E). Intriguingly, these data are in contrary to the known function of HSP70 and indicate that HSP70-TRAF6 complex is not the final effector for exosome-mediated hemolymph microbiota homeostasis. Therefore, we performed pull-down analysis based on TRAF6, it was found that TRAF6 could bind with Ecsit (Fig 6F and 6G and S2 Table). Given that HSP70 could bind to TRAF6, while TRAF6 also binds to Ecsit, we went on to use co-immunoprecipitation analysis to determine whether HSP70 could also directly bind to Ecsit. The results showed that HSP70 could not bind to Ecsit (Fig 6H), which suggests that TRAF6 formed separate complexes with HSP70 and Ecsit. Furthermore, we found that exosome-Vp could suppress the expression of HSP70 via exosomal miR-224, and the formation of TRAF6-HSP70 complex is inhibited due to the low expressed HSP70, then the released TRAF6 could further interact with Ecsit, resulting in the increased formation of TRAF6-Ecsit complex (Fig 6I). It was obvious that the expression of Ecsit significantly increased with the exosome-Vp treatment. All the above results in this section suggest that during the exosome-mediated regulatory process, interaction between HSP70 and TRAF6 is decreased, so that the released TRAF6 could form a complex with Ecsit.

## Role of the TRAF6-Ecsit complex in modulation of hemolymph microbiota homeostasis

The TRAF6-Ecsit complex is required for mitochondrial recruitment to phagosomes and is also involved in ROS production during anti-bacterial response [30]. Thus, when the expression of TRAF6 in the mitochondria of mud crabs treated with exosome-Vp was determined, an increased level of TRAF6 was observed (Fig 7A). Moreover, mROS level was also significantly increased in hemocytes of exosome-Vp treated mud crabs compared with controls (Fig 7B and 7C). These results indicate that released TRAF6 translocate to the mitochondria to form a complex with Ecsit, which then mediates mROS production. Ecsit is not only found in the mitochondria, but also in cytoplasm [31]. Thus, because TRAF6 also serves as an E3

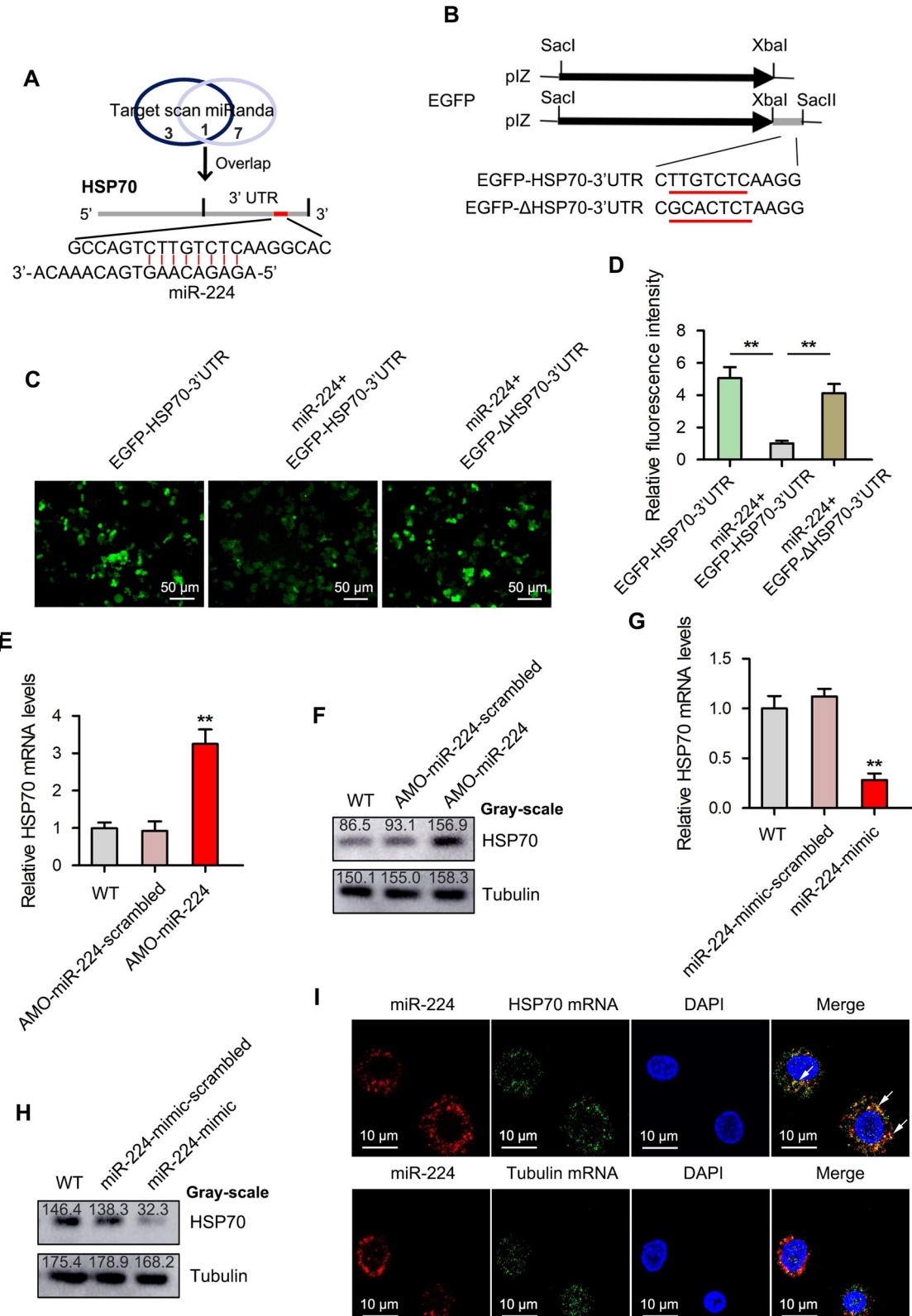

**Fig 4. HSP70 is the direct downstream target for miR-224 in mud crab. (A)** Target gene prediction of miR-224 using Targetscan and miRanda softwares. **(B)** Cloning of wild-type and mutated 3'UTRs of HSP70 into the pIZ-V5-EGFP plasmid. The sequences

targeted by miR-224 are underlined. (**C-D**) The direct interactions between miR-224 and HSP70 in insect cells. *Drosophila* S2 cells were co-transfected with miR-224 and/or the indicated plasmids for 48 h, followed by analysis of the relative fluorescence intensities. Data were presented relative to the value of miR-224+ EGFP-HSP70-3'UTR group, which was treated as standard "1". (**E-F**) The effect of miR-224 silencing on the expression level of HSP70 in mud crab post-injection with AMO-miR-224. The mRNA (**E**) and protein (**F**) levels were examined at 48 h post-injection. Gray-scale value quantification was conducted using Image J software. (**G-H**) The effect of miR-224 overexpression on the mRNA and protein levels of HSP70 in mud crabs. (**I**) The co-localization of miR-224 and HSP70 mRNA in mud crab hemocytes. miR-224 and HSP70 mRNA were determined with FAM-labeled miR-224 probe (red) and Cy3-labeled HSP70 mRNA probe (green), Cy3-labeled Tubulin mRNA probe (green) was used as a negative control, the arrows indicated co-localization. Experiments were performed in triplicates, with the data shown representing the mean ± s.d. (**, $p < 0.01$).

ubiquitin ligase [32], whether the binding of TRAF6 to Ecsit results in the ubiquitination of Ecsit requires further investigation. Based on this, the effect of TRAF6 silencing on Ecsit ubiquitination was determined. As shown in Fig 7D, TRAF6 knockdown resulted in significant decrease in Ecsit ubiquitination, which suggest that TRAF6 was the ubiquitin ligase of Ecsit. Besides, it was found that the ubiquitination of Ecsit was significantly increased when the mud crabs were treated with exosome-Vp (Fig 7E). It has been reported that ubiquitination is a signal for nuclear translocation [33]. For this reason, when the nuclear translocation of Ecsit was determined using Western blot analysis, the results revealed an increased in the protein level of Ecsit in nuclear extracts of hemocytes from mud crabs injected with exosome-Vp (Fig 7F). Furthermore, the localization of Ecsit was confirmed by immunofluorescence microscopy technique using mouse anti-Ecsit antibody. The results indicated co-staining of Ecsit with DAPI in hemocytes nuclei (Fig 7G).

To explore the effect of Ecsit translocation to the nucleus, dual-luciferase reporter assay was carried out in S2 cells. The results showed that the overexpression of Ecsit resulted in significant activation of ALF1, ALF4 and ALF5 transcription (Fig 7H). The role of the TRAF6-Ecsit complex in the modulation of hemolymph microbiota homeostasis in mud crabs was then explored using 16S rDNA sequencing after knockdown of TRAF6 and Ecsit, respectively. The results revealed that the mediation of hemolymph microbiota homeostasis by exosome-Vp was disrupted upon silencing of TRAF6 or Ecsit (Fig 7I and 7J), which indicates that the TRAF6-Ecsit complex is required for exosome-mediated hemolymph microbiota homeostasis. Furthermore, to evaluate the effect of TRAF6-Ecsit complex during exosome-mediated pathogenic bacteria resistance in mud crab, the results showed that exosome-VP-mediated anti-bacterial function was significantly reduced when TRAF6 or Ecsit was silenced and therefore affecting the mud crab mortality rate (Fig 7K).

Taken together, the findings in this study indicate that during *V. parahaemolyticus* infection, there is more packaging of miR-224 in the mud crab exosomes. This increased uptake of exosomal miR-224 resulted in HSP70 suppression, which causes disruption of the HSP70-TRAF6 complex in recipient hemocytes, thereby releasing TRAF6 to interact with Ecsit in mitochondria to regulate mROS production. TRAF6 also mediates Ecsit ubiquitination and nuclear translocation to facilitate the transcription of ALFs, which affect hemolymph microbiota homeostasis in response to pathogens infection (Fig 8).

## Discussion

Exosomes are small bioactive membrane-enclosed vesicles derived from the fusion of multivesicular bodies (MVBs) with the plasma membrane that promote intercellular communication [34]. There is growing evidence that exosomes are involved in the regulation of pathogen infection and immune response of the host [35]. For instance, exosomes released from *Mycobacterium avium* (*M. avium*)-infected macrophages contain Glycopeptidolipids (the major cell wall constituent of *M. avium*) that are transferred to uninfected macrophages to stimulate a

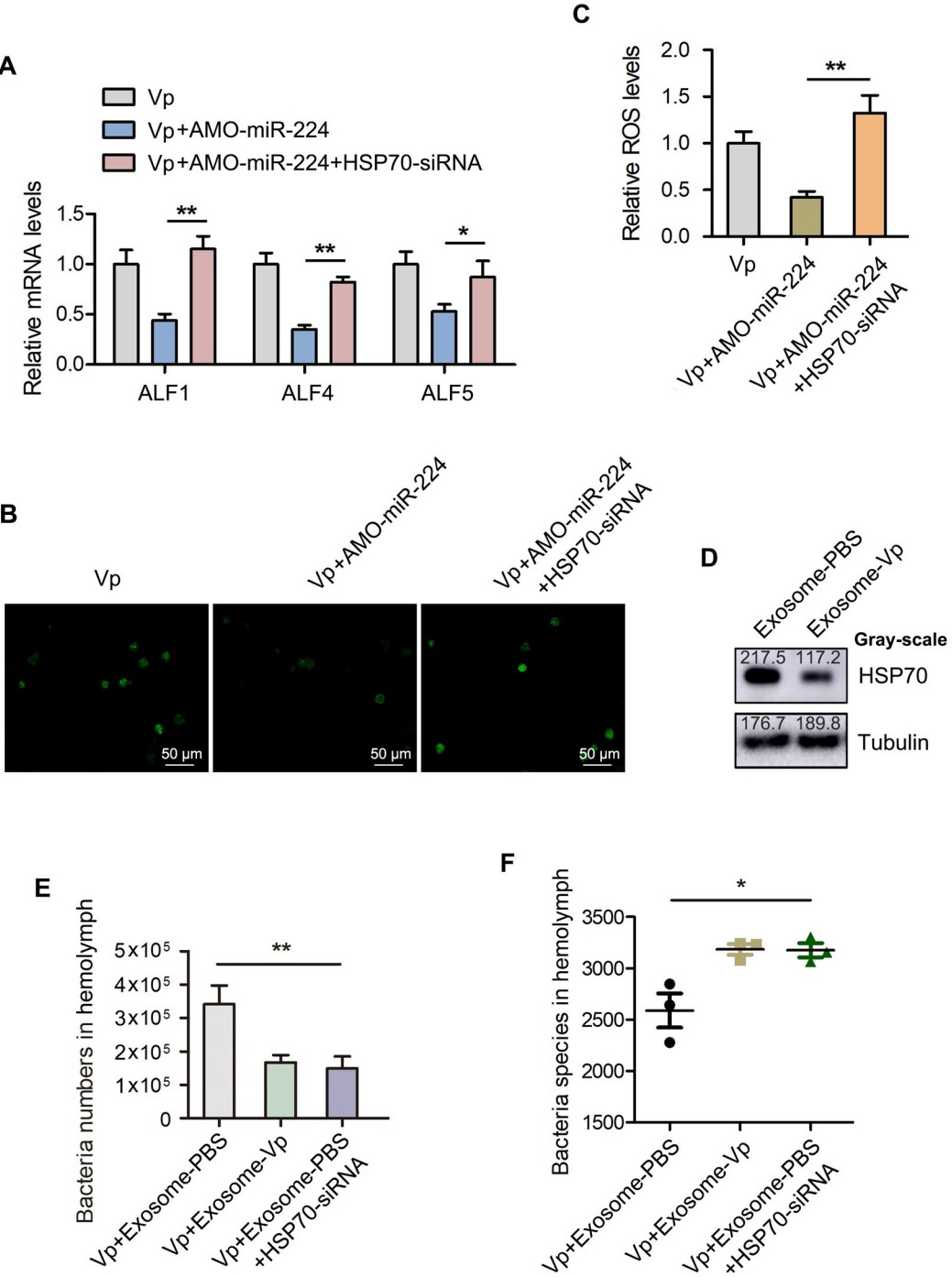

**Fig 5. Role of HSP70 in exosomal miR-224-mediated hemolymph microbiota homeostasis. (A)** The participation of HSP70 in miR-224-mediated ALFs regulation in mud crabs. AMO-miR-224 was co-injected with HSP70-siRNA into *V. parahaemolyticus*-challenged mud crabs, followed by analysis of the expression levels of ALFs using qPCR. Data were presented relative to the value of Vp group, which was treated as standard "1". **(B-C)** The involvement of HSP70 in miR-224-mediated ROS production. The level of ROS in mud crab hemocytes was determined using fluorescence microscopy, Scale bar, 50 μm **(B)** and microplate reader (Data were presented relative to the value of Vp group, which was treated as standard "1") **(C)**. **(D)** The effect of the indicated exosomes on HSP70 expression. Isolated exosomes from mud crabs treated with PBS and *V. parahaemolyticus* were injected into mud crabs for 48 h, followed by determination of HSP70 protein level using Western blot analysis, tubulin was used as an internal reference. **(E-F)** The effect of HSP70 silencing on exosome-mediated hemolymph microbiota homeostasis. Hemolymph was collected from mud crabs with the indicated treatments and subjected to 16S rDNA sequencing, then the bacterial cell count **(E)** and species **(F)** were analyzed. The data

of Vp+Exosome-PBS and Vp+Exosome-Vp groups were from Fig 2D. All the data are the average from at least three independent experiments, mean ± s.d. (*, $p < 0.05$; **, $p < 0.01$).

pro-inflammatory response dependent on Toll like receptor (TLR) 2, TLR4, and MyD88 [36]. Similarly, it has been reported that exosomes released from macrophages infected with *Mycobacterium tuberculosis*, *Salmonella typhimurium* and *Mycobacterium bovis* could stimulate Tumor necrosis factor alpha and interleukin-12 production in mice [37]. Currently, studies on exosome-bacterial interaction during infections have mainly been carried out in higher organisms. However, the role of exosomes in anti-bacterial immunity in invertebrates has largely not been explored. Moreover, most of these studies are related to inflammation regulation [38], while the involvement of exosomes in anti-bacterial immunity has never been addressed from the perspective of hemolymph microbiota homeostasis particularly in crustacean. In the current study, we found that exosomes released from *V. parahaemolyticus*-infected mud crabs could reduce crab mortality due to bacterial infection by maintaining hemolymph microbiota homeostasis. This is the first time it has been demonstrated that exosomes play a role during anti-bacterial immunity of invertebrates, and also shows the involvement of exosomes mediation in hemolymph microbiota homeostasis during response to pathogens infection.

One of the typical features of exosomes is the packaging of large numbers of nucleic acids, including miRNA, mRNA, mtDNA, piRNA, lncRNA, rRNA, snRNA and tRNA [39]. Given that miRNAs are the most abundant RNA species in exosomes, it has been reported that the molecular composition of the miRNA cargo carried by exosomes can be affected by external signals such as oxidative stress and pathogens infection, which reflects the physiological or pathological state of donor cells [40]. Our previous study revealed that miR-137 and miR-7847 were less packaged in mud crab exosomes after White spot syndrome virus challenge, which resulted in the activation of apoptosis induce factor and eventually the induction of apoptosis and suppression of viral infection in recipient mud crab hemocytes [41]. Besides, miR-145, miR-199a, miR-221 and Let-7f that are assembled in exosomes can directly bind to the genomic RNA of Hepatitis C virus to inhibit viral replication in umbilical cord mesenchymal stem cells [42]. In addition, exosomal miR-21 and miR-29a regulate gene expression in HEK293 cells as well as function as ligands that bind with TLRs to activate relevant immune pathways in recipient cells [43]. Due to their diverse regulatory roles, exosomal miRNAs have been shown in most studies to be crucial regulators of host-pathogen interactions, mainly studies involving viral infections [44]. Thus far, the role of exosomal miRNAs in bacterial infections still remain unexplored, especially in invertebrates. The current study reveals that during *V. parahaemolyticus* infection in mud crabs, miR-224 was more packaged in exosomes, which resulted in the suppression of HSP70, and eventually affected hemolymph microbiota homeostasis by regulating the levels of ROS and ALFs expression to help clear the infection. This observation reveals a novel regulatory mechanism that shows the role of exosomal miRNAs during innate immune response in invertebrates.

Most of the miR-224-associated studies have been conducted in human cancer cells, with miR-224 reported to promote the expression of tumor invasion-associated proteins p-PAK4 and MMP-9 by directly targeting HOXD10 [45]. It has also been shown that miR-224 can be packaged into exosomes released by hepatocellular carcinoma to regulate cell proliferation and invasion by targeting glycine *N*-methyltransferase [46]. While the role of miR-224 in invertebrates had remained elusive, results from this current study reveals that miR-224 could target HSP70 to disrupt the HSP70-TRAF6 complex or its formation. As an evolutionarily conserved protein, HSP70 plays an essential role during the regulation of cell growth, senescence and apoptosis [47]. Studies have shown that HSP70 could inhibit LPS-induced NF-κB activation

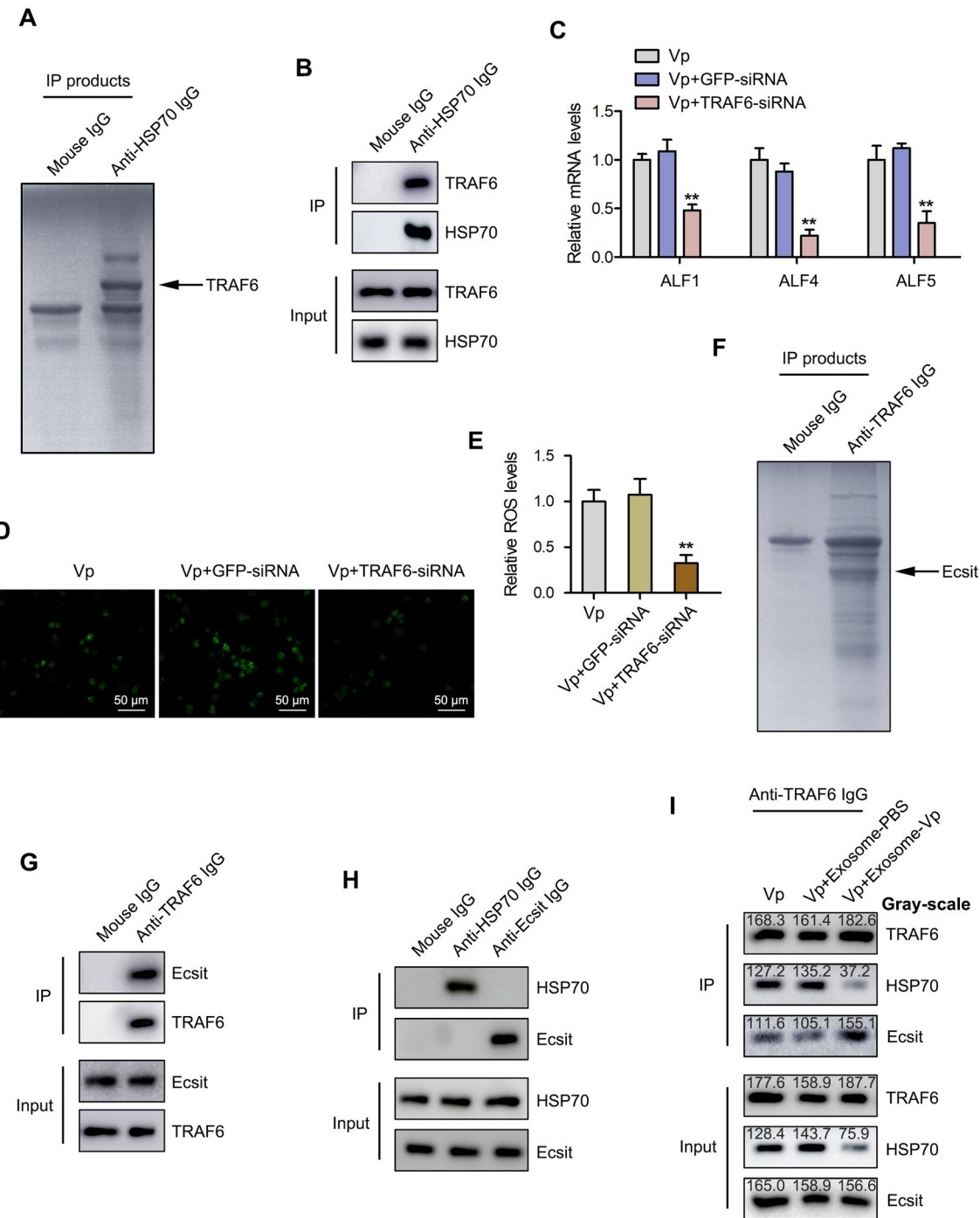

**Fig 6. miR-224-mediated suppression of HSP70 results in disruption of the HSP70-TRAF6 complex and TRAF6-Ecsit complex formation. (A)** Identification of proteins that bind to HSP70. Mud crab hemocytes lysates were subjected to Co-immunoprecipitation (Co-IP) assay using anti-HSP70 IgG, followed by separation using SDS-PAGE and identification of the proteins by mass spectrometry. **(B)** Interactions between HSP70 and TRAF6 in mud crab, the cell lysates were analyzed using Co-IP with anti-HSP70 IgG, then the IP products were subjected to Western blot assay to detect TRAF6. **(C)** The effect of TRAF6 silencing on ALFs regulation. Mud crabs were injected with TRAF6-siRNA or GFP-siRNA for 48 h, followed by analysis of ALFs expression using qPCR. Data were presented relative to the value of Vp group, which was treated as standard "1". **(D-E)** Effect of TRAF6 silencing on ROS production in mud crabs. The level of ROS in mud crab hemocytes was analyzed using fluorescence microscopy, Scale bar, 50 μm **(D)** and microplate reader (Data were presented relative to the value of Vp group, which was treated as standard "1") **(E)**. **(F)** Identification of proteins that bind to TRAF6. Mud crab hemocytes lysates were subjected to Co-IP assay using anti-TRAF6 IgG, followed by separation using SDS-PAGE and identification of the proteins by mass spectrometry. **(G)** The interaction between

TRAF6 and Ecsit in mud crabs. The cell lysates were analyzed using Co-IP with anti-TRAF6 IgG, then the IP products were subjected to Western blot assay to detect Ecsit. **(H)** The interaction between HSP70 and Ecsit in mud crabs. Cell lysates were subjected to Co-IP analysis with anti-HSP70 IgG and anti-Ecsit IgG, followed by Western blot analysis using the indicated antibodies. **(I)** The interactions between HSP70 and TRAF6, TRAF6 and Ecsit in mud crabs after the indicated treatments. Hemocytes lysates were collected from mud crabs challenged with Vp, Vp+ Exosome-PBS and Vp+ Exosome-Vp, followed by Co-IP assay using anti-TRAF6 IgG, then the IP products were subjected to Western blot assay to detect HSP70 and Ecsit. Data shown represent the mean ± s.d. for triplicate assays (**, $p<0.01$).

by interacting with TRAF6 to prevent its ubiquitination, which eventually suppresses the production of mediators of inflammation [48]. Similarly, our current data show that HSP70 could bind with TRAF6 to affect its function in mud crab hemocytes, while the release of TRAF6 from disruption of the HSP70-TRAF6 complex allows TRAF6 to complex with Ecsit. The TRAF6-Ecsit complex is required for mitochondrial recruitment to phagosomes, hence, disruption of the TRAF6-Ecsit complex would severely dampen ROS production and therefore increase susceptibility to bacterial infection [30]. Both TRAF6 and Ecsit have also been reported to regulate the expression of AMPs during bacterial infection in marine crustaceans [49,50]. In the present study, it was observed that TRAF6 cooperates with Ecsit to regulate mROS and the expression of ALFs in mitochondria and nuclei, respectively, which further affects hemolymph microbiota homeostasis in response to bacterial infection in mud crabs. The present study therefore provides novel insights into how invertebrates mount resistance to pathogenic microbial infections.

## Materials and methods

### Ethics statement

The mud crabs used in this study were purchased from a local crab farm (Niutianyang, Shantou, Guangdong, China), and processed according to the Regulations for the Administration of Affairs Concerning Experimental Animals established by the Guangdong Provincial Department of Science and Technology on the Use and Care of Animals. The relevant studies did not involve endangered or protected species and therefore no specific permits were required for the described field studies.

### Mud crab culture, *V. parahaemolyticus* challenge and mortality analysis

Healthy mud crabs, approximately 50 g each, were acclimated to laboratory conditions in water with 8‰ salinity at 25°C for a week before further processed. For pathogen challenge, 200 μL *V. parahaemolyticus* ($1\times10^7$ *cfu*/mL) was injected into the base of the fourth leg of each crab, 10 mM PBS (PH = 7.4) was used as control. At different time post-infection, hemolymph was collected from three randomly chosen crabs per group for further investigation [27]. For mud crab mortality assays, 10 crabs/per treatment were used, 200 uL of the indicated exosome solution ($1\times10^8$ vesicles/mL) was injected into the base of the fourth leg of each crab according to our previous study [41]. The cumulative mortality was examined daily. Every experiment, including pathogens infection and mortality assay, was biologically repeated for three times. These data were used for statistical analysis.

### Isolation and analysis of exosomes

For exosomes isolation, 50 mL hemolymph from mud crabs were separated, after centrifuged at $300 \times g$ for 5 min, to collect the supernatant. Next, supernatants were subjected to ultracentrifugation, followed by sucrose density-gradient centrifugation and filtrated through filters (pore size of 0.22 μm). The obtained exosomes were observed by Philips CM120 BioTwin

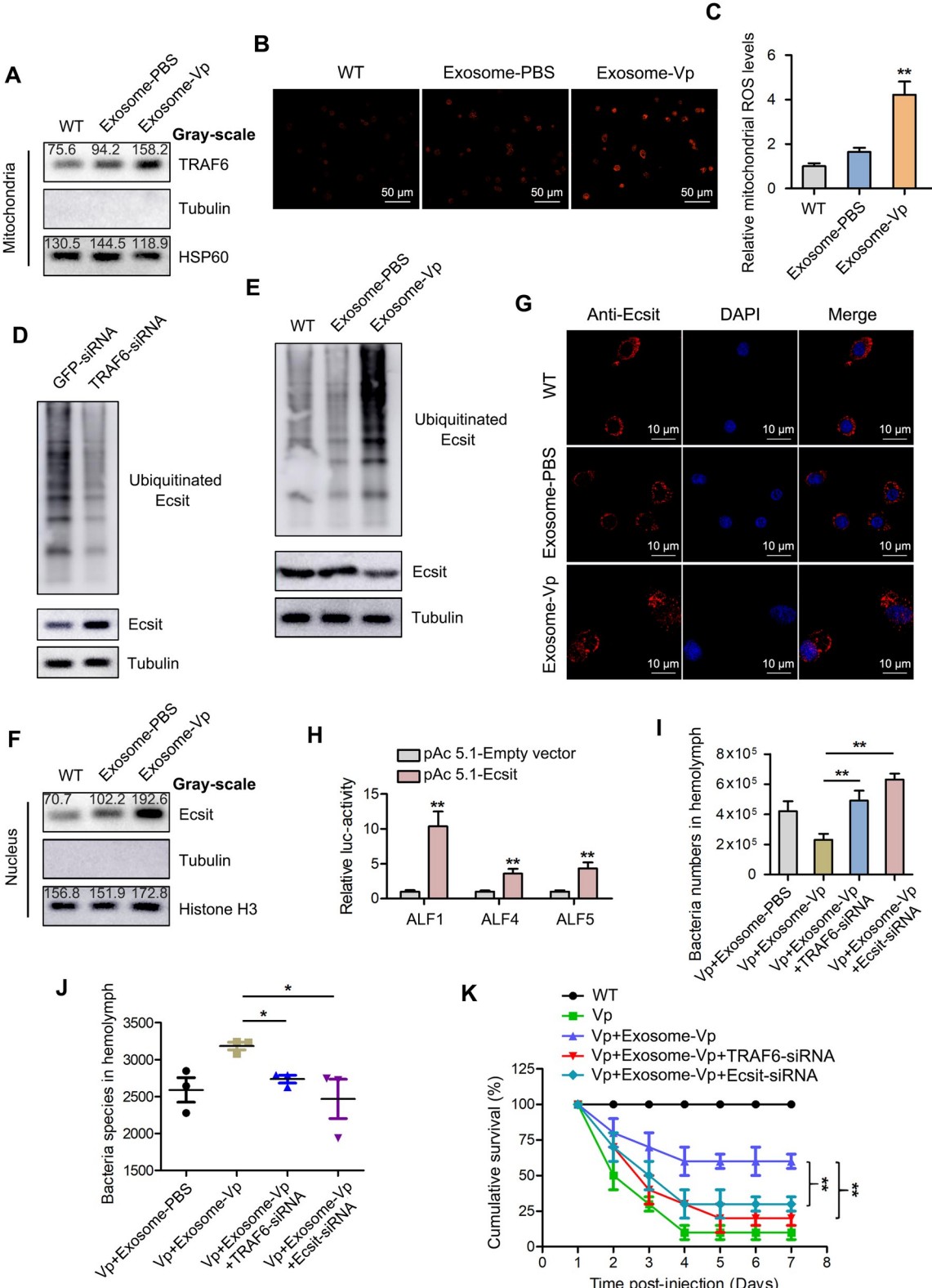

**Fig 7. TRAF6-Ecsit complex mediates hemolymph microbiota homeostasis. (A)** The effect of the indicated exosomes on the protein level of TRAF6 in mitochondria. Mud crabs were treated with either Exosome-PBS or Exosome-Vp for 48 h, then the mitochondria were isolated and subjected to Western blot assay to detect TRAF6. Tubulin and HSP60 were used as negative indicator and positive

indicator to evaluate the purity of the isolated mitochondria, respectively. **(B-C)** The effect of the indicated exosomes on mROS production. The mROS level in mud crab hemocytes was determined using fluorescence microscopy, Scale bar, 50 μm **(B)** and microplate reader (Data were presented relative to the value of WT group, which was treated as standard "1") **(C)**. **(D)** The effect of TRAF6 silencing on the expression and ubiquitination levels of Ecsit. Mud crabs were treated with TRAF6-siRNA for 48 h, followed by the detection of total and ubiquitinated Ecsit through Western blot assay. **(E)** The effect of the indicated exosomes on the expression and ubiquitination levels of Ecsit. Mud crabs were treated with Exosome-PBS or Exosome-Vp for 48 h, followed by the detection of total and ubiquitinated Ecsit through Western blot assay. **(F)** The protein level of Ecsit in mud crab hemocytes nuclei after treatment with the indicated exosomes was determined by Western blot analysis. Tubulin and Histone H3 were used to evaluate the purity of the isolated nuclei. **(G)** The localization of Ecsit in mud crab hemocytes after treated with the indicated exosomes was determined using immunofluorescence assay with mouse anti-Ecsit antibody, Scale bar, 10 μm. **(H)** The effect of Ecsit overexpression on the transcription of ALFs. pGL3-Basic and renilla luciferase activities serve as internal reference, Data were presented relative to the value of pAc 5.1-Empty vector group, which was treated as standard "1". **(I-J)** The effects of TRAF6 or Ecsit silencing on exosome-mediated hemolymph microbiota homeostasis. Hemolymph was collected from mud crabs after the indicated treatments and was used to determine bacteria number **(I)** and species **(J)**. The data of Vp+Exosome-PBS and Vp+Exosome-Vp groups were from Fig 2D. **(K)** Effects of TRAF6 or Ecsit silencing on exosome-VP-mediated anti-bacterial function in mud crab. The specific treatments are shown on the top and the mortality was examined daily. Data shown represent that of three independent experiments (*, $p<0.05$; **, $p<0.01$).

transmission electron microscope (FEI Company, USA), while the quantity and size of the exosomes were measured by Nano-Sight NS300 (Malvern Instruments Ltd, UK).

## Exosomal miRNAs sequencing

Exosomal miRNAs were identified through miRNA sequencing based on Illumina platform by a commercial company (Biomarker Technologies, Beijing, China), the reference genome used was the genome of mud crab, and miRbase database used for miRNA identification was Metazoa database. The original sequencing data were uploaded to NCBI BioProject database with accession number PRJNA600674, hierarchical clustering analysis of the differential expressed miRNAs was conducted by Cluster 3.0 and TreeView software.

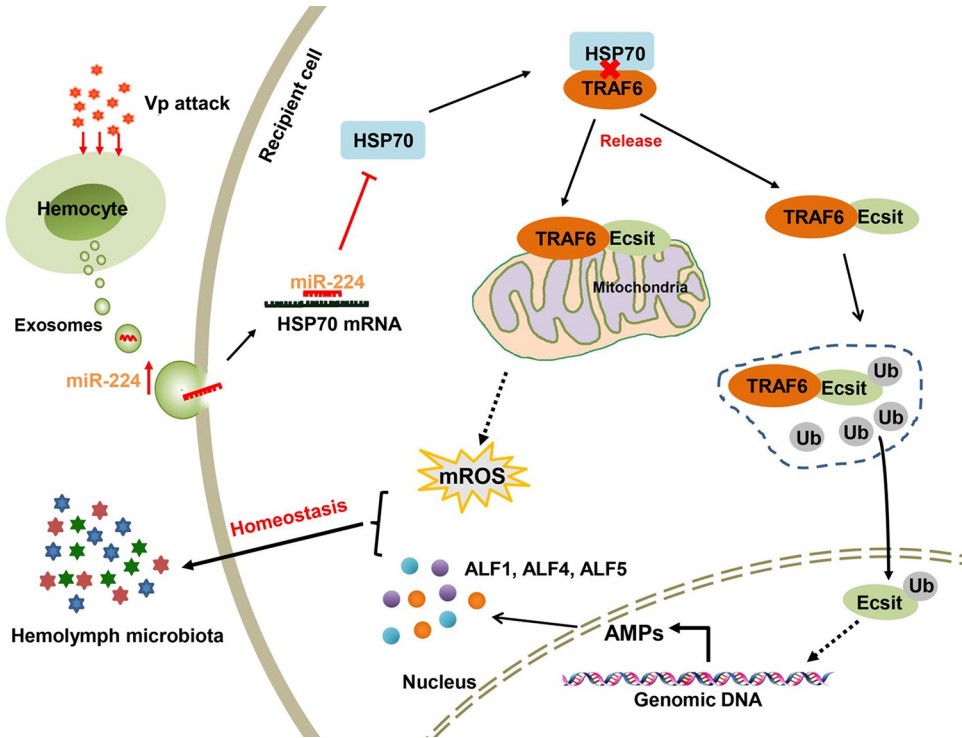

**Fig 8. Proposed schematic diagram for exosomal miR-224-mediated hemolymph microbiota homeostasis during** ***V. parahaemolyticus*** **infection in mud crabs.**

### Prediction of target genes

The target genes of miR-224 were predicted by a commercial company (BioMarker, Beijing, China) using Targetscan and miRanda software (http://www.targetscan.org; http://www.microrna.org/). The overlapped target genes predicted by the two algorithms were treated as the candidate target gene.

### Overexpression and silencing of miR-224 in mud crabs

The miRNA mimics and anti-microRNA oligonucleotides (AMOs) of miR-224 were injected at 30 μg/crab for 48 h to overexpress and knockdown miR-224 in mud crab respectively. miR-224 mimic (5'-AGAGACAAGTGAC**A**AA**C**A-3') and AMO-miR-224 (5'-TGTTTGTCACTTG**T**CT**C**T-3') were modified with 2'-*O*-methyl (OME) (bold letters) and the remaining nucleotides phosphorothioated. All oligonucleotides were synthesized by Sangon Biotech (Shanghai, China).

### Cell culture, transfection, and fluorescence assays

The *Drosophila* Schneider 2 (S2) cells were cultured at 27˚C with Express Five serum-free medium (SFM) (Invitrogen, USA). The EGFP-HSP70-3'UTR or mutant plasmids (100 ng/well) and the synthesized miR-224 (50 nM/well) were co-transfected into S2 cells using Cellfectin II Reagent (Invitrogen, USA) according to the manufacturer's protocol. At 48 h post-transfection, the EGFP fluorescence in S2 cells were observed under an inverted fluorescence microscope (Leica, Germany) and measured by a Flex Station II microplate reader (Molecular Devices, USA) at 490/ 510 nm of excitation/emission (Ex/Em).

### RNA interference assay

Based on the nucleotide sequence of HSP70, TRAF6 and Ecsit, the specific siRNAs targeting these genes were designed, i.e., HSP70-siRNA (5'- UCUUCAUAAGCACCAUAGAGGAGUU-3'), TRAF6-siRNA (5'-GCUUCUCCCAGCUUGCAAUUU-3') and Ecsit-siRNA (5'-CCCUGUACUCUUCACACAAUU-3'). The siRNAs were synthesized using *in vitro* Transcription T7 Kit (TaKaRa, Dalian, China) according to the manufacturer's instructions. Next, 50 μg of each siRNA was injected into each mud crab. At different time points post injection, three mud crabs were randomly selected for each treatment and stored for later use.

### Quantitative real-time PCR of mRNA

Quantitative real-time PCR was conducted to quantify the mRNA levels using Premix Ex Taq (Takara, Japan). Total RNA was extracted from mud crab hemocytes, followed by first-strand cDNA synthesis using PrimeScript RT Reagent Kit (Takara, Japan). Primers ALF1-F (5'-AACTCATCACGGAGAATAACGC-3'), ALF1-R (5'- CTTCCTCGTTGTTTTCACCCTC-3'); ALF2-F (5'-TGTCGCTCAGGGACTCATCAC-3'), ALF2-R (5'-GGAGATCACGGGAGAGTGAATG-3'); ALF3-F (5'- GAACGGACTCATCACACAGCAG-3'), ALF3-R (5'-CACTTCCTTGTTCTCTTCGCTC-3'); ALF4-F (5'-CACTACTGTGTCCTGAGCCGC-3'), ALF4-R (5'- GTCCTCGCCTTACAATCTTCTG-3'); ALF5-F (5'-CTTGAAGGGACGAGGTGATGAG-3'), ALF5-R (5'-TGACCAGCCCATTCGCTACAG-3'); ALF6-F (5'-ACAGGGCTATCGCAGACTTCG-3'), ALF6-R (5'-GCACCTCTTTGGCACACTATTTG-3') were used to quantify transcripts levels of ALFs. β-actin (F, 5'-GCGGCAGTGGTCATCTCCT-3' and R, 5'-GCCCTTCCTCACGCTATCCT-3') was used as the internal control. Relative fold change was determined using the $2^{-\Delta\Delta Ct}$ algorithm [51].

## Quantification of miRNA with real-time PCR

Total RNA was extracted from hemocytes or exosomes using MagMAX mirVana Total RNA Isolation Kit (Thermo Fisher Scientific, USA), then, PrimeScript II 1st Strand cDNA Synthesis Kit (Takara, Japan) was used to conduct first-strand cDNA synthesis with miR-224-primer (5'-GTCGTATCCAGTGCAGGGTCCGAGGTCACTGGATACGACTGTTTGTC-3'). After that, Real-time PCR was carried out using the Premix Ex Taq (Takara, Japan) to quantify the level of miR-224 with miR-224-F (5'-CGCCGAGAGACAAGTGAC-3') and miR-224-R (5'-TGCAGGGTCCGAGGTCACTG-3'). U6 (F, 5'-CTCGCTTCGGCAGCACA-3' and R, 5'-AACGCTTCACGAATTTGCGT-3') was used as an internal reference. Relative fold change was determined using the $2^{-\Delta\Delta Ct}$ algorithm [51].

## Hemolymph bacteria counting and sequencing

Mud crab hemolymph collected from each group after specific treatments was passed through 5-μm-pore-size mesh membrane, after that, the filtrate was filtered through 0.2-μm-pore-size mesh membrane. Then, microbial cells on the 0.2-μm-pore-size mesh membrane were stained with SYBR Green I solution (1:40 v/v SYBR Green I in 1× Tris EDTA buffer). Each treatment was consisted of 5 crabs and there were three biological replicates for each group, meaning that each group represented 15 individuals. Next, the bacteria number were counted under 100× magnification using a fluorescence microscope (Axio Imager M2, Zeiss, Germany) ([27] for details). For 16S rDNA genes sequencing, the total genome DNA of hemolymph microbiota was extracted using QIAamp PowerFecal DNA Kit (Qiagen, Germany). All samples were sequenced on Illumina Nova platform by a commercial company (Novogene, Beijing, China) by amplifying the V4 region of the 16S rDNA gene, the data were uploaded to NCBI BioProject database (accession number PRJNA669103).

## Cellular and mitochondrial ROS measurement

5 mL hemolymph from three randomly chosen mud crabs per group were drawn into tubes containing ACD anticoagulant buffer, and then centrifuged immediately at 800 ×$g$ for 20 min at 4˚C to isolate the hemocytes. Next, flow cytometry method was used to measure cellular ROS level with a ROS Assay Kit (Beyotime Biotechnology, China). For mitochondrial ROS measurement, the ROS intensity was analyzed by MitoSOX Red Mitochondrial Superoxide Indicator (Invitrogen, USA). The fluorescence in hemocytes was observed using an inverted fluorescence microscope (Leica, Germany) and measured by a Flex Station II microplate reader (Molecular Devices, USA).

## Immunoprecipitation analysis

Mud crab hemocytes were collected and lysed with ice-cold cell lysis buffer (Beyotime, China). Followed by mixed with Protein G-agarose beads (Invitrogen, USA) and incubated with HSP70 or TRAF6 antibody at 4˚C overnight. After being washed for three times with ice-cold lysis buffer, the immuno-complexes were eluted by 0.1M glycine and then subjected to SDS-PAGE or LC-MS/MS analysis.

## Western blot analysis

Mud crab hemocytes were collected and homogenized with RIPA buffer (Beyotime Biotechnology, China) containing 1 mM phenylmethanesulfonyl fluoride (PMSF), then the cell extracts were mixed with 5 × SDS sample buffer and separated by 12% SDS-polyacrylamide gel electrophoresis. After that the proteins were transferred onto a nitrocellulose membrane (Millipore, USA). Subsequently, the membrane was blocked and further incubated with

appropriate primary antibody at 4˚C. Followed by washed with TBST and incubated with horseradish peroxidase-conjugated secondary antibody (Bio-Rad, USA) for subsequent detection by ECL substrate (Thermo Scientific, USA). The primary antibody used is mouse polyclonal antibody prepared in our lab.

## Statistical analysis

All data were subjected to one-way ANOVA analysis using Origin Pro8.0, with $P < 0.01$ considered as statistically significant. All experiments were carried out in triplicates and repeated for three biological replicates.

## Supporting information

**S1 Fig. The affinity analysis of exosomes to mud crab hemocytes.** The indicated exosomes were labeled with Dio (green) for 6 h, after which the labeled exosomes were injected to the mud crab and then hemocytes were isolated and analyzed by flow cytometry. Experiments were performed in triplicates, with the data shown representing the mean ± s.d. (*, p<0.05; **, p<0.01).
(TIF)

**S1 Table. LC-MS/MS data of IP products by HSP70 antibody.**
(DOCX)

**S2 Table. LC-MS/MS data of IP products by TRAF6 antibody.**
(DOCX)

## Acknowledgments

We sincerely thank professor Xiaobo Zhang of Zhejiang University for providing *Drosophila* S2 cells and pIZ-V5-EGFP vector. We also thank Tran Ngoc Tuan of Shantou University for editing the paper.

## Author Contributions

**Data curation:** Yi Gong, Yueling Zhang, Shengkang Li.

**Funding acquisition:** Yi Gong, Yueling Zhang, Shengkang Li.

**Investigation:** Yi Gong, Xiaoyuan Wei, Wanwei Sun, Xin Ren, Jiao Chen.

**Methodology:** Hongyu Ma, Kok-Gan Chan, Yueling Zhang, Shengkang Li.

**Project administration:** Yueling Zhang, Shengkang Li.

**Validation:** Yi Gong, Xiaoyuan Wei, Wanwei Sun, Xin Ren, Jiao Chen.

**Writing – original draft:** Yi Gong.

**Writing – review & editing:** Jude Juventus Aweya, Shengkang Li.

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
