## [Decision Letter · Decision Letter 0]

18 Jun 2021

Dear Dr. Li,

Thank you very much for submitting your manuscript "Exosomal miR-224 contributes to hemolymph microbiota homeostasis during bacterial infection in crustacean" for consideration at PLOS Pathogens. As with all papers reviewed by the journal, your manuscript was reviewed by members of the editorial board and by several independent reviewers. The reviewers appreciated the attention to an important topic. Based on the reviews, we are likely to accept this manuscript for publication, providing that you modify the manuscript according to the review recommendations.

In addition to the corrections required by reviewers, I would like to draw attention to the fact that throughout the manuscript, including the title, it is defended and conveyed to readers that the modulation of pathways triggered by miR-224-containing exosomes  is carried out directly in the microbiome and that is how *V. parahaemolyticus* infection is controlled, or at least contributes to control of the infection. However, without discarding this possibility, there is no data in the manuscript to indicate this direct relationship between the triggered pathways and their effect on the microbiota, and it cannot be ruled out that the increase in ROS and ALFs are acting instead on *V. parahaemolyticus* and the elimination or decrease of the pathogen, is what leads to restauration/homeostasis of the microbiota. Authors must present results that clearly show that the role of ROS and ALFs is directly in the microbiota or, alternatively, discuss both possibilities and correct the title and the most striking sentences related to this point in the article.

Sincerely,

Nuno Miguel Simões dos Santos

Guest Editor

PLOS Pathogens

Karla Satchell

Section Editor

PLOS Pathogens

Kasturi Haldar

Editor-in-Chief

PLOS Pathogens

orcid.org/0000-0001-5065-158X

Michael Malim

Editor-in-Chief

PLOS Pathogens

orcid.org/0000-0002-7699-2064

Reviewer Comments (if any, and for reference):

Reviewer's Responses to Questions

**Part I - Summary**

Reviewer #1: Gong et al propose a manuscript entitled “Exosomal miR-224 contributes to hemolymph microbiota homeostasis during bacterial infection in crustacean”. The authors assess the role of extracellular vesicles released by V. parahaemolyticus infected mud crabs into the hemolymph. They claim that these extracellular vesicles have a protective role against Vp infection reducing crab mortality by maintaining the homeostasis of hemolymph microbiota. They focus on miR-244 which they found enriched in extracellular vesicles, and propose that it targets HSP70 downregulating its expression, which allows TRAF6 to interact with Ecsit ultimately regulating the expression of mROS and ALFs.

The data reported is relevant and of potential broad interest, in addition it may impact different fields of research. The manuscript has been previously submitted for publication in Plos Path. In this new version the authors assess the referee comments and performed additional experiments. However, the manuscript still lacks important details which perturb the proper appreciation of the data. This is a very extensive study but the authors fail to properly present the data and argument the findings. As an example, all the new performed experiments were added as supplementary data, with no effort to integrate some of them (which are of critical importance, as Fig S3) in the main manuscript. The new data and its implications are not exploited to improve the findings.

Reviewer #2: The authors were able to address many of the concerns that I raised in the review comments. However, there are still several issues remaining to be addressed before the current manuscript is accepted for publication.

**Part II – Major Issues: Key Experiments Required for Acceptance**

Reviewer #1: The authors need to look at all data and provide the controls missing, the quantifications, the details of the experiments and better images, and think about how they need to present them to convince the readers. The results section appears as an accumulation of observations and lack well supported conclusions, hypothesis, analysis in an integrative manner.

In general, the figure legends need to be more detailed: the kind of experiments should be mentioned, the number of animals, the number of independent experiments. The authors made the effort to add some of these in the material and methods section, but reduced info should also appear in Figure legends.

The quantification (all the quantifications) should be performed on at least 3 independent experiments and should appear as a graph including all the quantifications. Quantify the single experiment that is shown is not enough and only has a limited interest.

Abstract

The abstract needs to be revised concerning the language and the content. For example, the first sentence is hard to follow. Actually, the Author summary is much clear than the scientific abstract itself. I strongly recommend revision.

FIG1

Lines 157-159: The data does not demonstrate what is claimed. Instead the data presented shows that extracellular vesicles derived from Vp-infected mud crabs have a protective effect on the mortality and proliferation of bacteria in the hemolymph, which may suggest a role in microbiota homeostasis and immune response.

A;B, C show that there is not major differences between Vp- or PBS-derived exosomes. The authors need to state this kind of conclusions.

In the additional data presented in S1 there is no indication on the number of experiments.

In F, why not counting the bacteria after plating. Counting under the microscope is a time-consuming way and is not as accurate. What fraction of the hemolymph was counted? From how many crabs?

FIG2

Line 185: The data do not show definitively that exosomes modulate pathways that maintain the homeostasis of hemolymph microbiota by regulating the levels of mROS and ALFs…. This is a possibility. The statement needs to be down modulated.

Images in A are of low quality. They do not add something interesting.

In C, readers need to know the number of experiments and the number of crabs, this should be added to the legend.

In F, the authors should discuss why some species that are that are highly present (red) upon Vp infection, almost disappear in the presence of both Vp- and PBS-derived exosomes.

FIG3

In B, why do the authors analyse the expression of ALF1, given that in Fig2 they show that ALF4 is much more up-regulated? Data are presented relative to what? Housekeeping expression? What mimic-NC stands for?

It seems that some data presented in I, are the same that are shown in 2F. If this is the case, this should be clearly stated. Any other similar situations in the manuscript should also be stated.

FiG4

In D, data are presented relative to what???

In E,F,G and H if quantifications shown as numbers in F and H are included in the graphs, the authors do not need to add them on the images of the gels… all those number are misleading!

In I, the image needs to be accompanied by a quantification (through Pearson’s correlative coefficient or other that can measure co-localization). Data from the control added in this new version as supplementary should be added here, with corresponding quantification, of course. The reader need to know how much miR-224 colocalizes with HSP70 mRNA and with tubulin mRNA.

FIG 5 and 6

Microscopy images showing mROS are of low quality.

Western blots need to be quantified from at least 3 independent experiments and data shown as a graph, number do not need to appear on the images.

Figure legends need to be carefully completed, as mentioned before. The important information and important details should be easily found.

FIG7

This figure should include data included in S3, this is a very nice piece of data that needs to exploited and shed to light in the manuscript.

Figure legend is poor, need to complete.

Reviewer #2: 1. The authors have attempted to address the questions of exosomal quality control that were raised during review. According to the definition of ISEV, exosomes are defined by the presence of two positive markers and a negative marker, such as GM-130, in addition to TEM and NTA. The authors should add MG-130 in the WB as a negative control (Fig. 1C).

2. In line 145, the authors stated that exosome-vp and exosome-PBS possess similar affinity to the recipient hemocytes. However, the Fig S1 shows the binding percentage not affinity of exosome-vp and exosome-PBS to hemocytes. You can’t measure affinity with a heterogenous population of exosomes for binding to cells. The authors should correct it.

3. The authors acclaimed in multiple statements in the manuscript that miR-224 was upregulated or highly expressed in exosomal preparations without providing any descriptions on how the exosomes were and miR-224 were quantified. The authors should provide detailed description on how miR-224 was quantified in exosome-vp, such as if mass of exo-vp and exo-PBS (Fig. 3D) was used as a control or other exosomal markers were used as control.

4. The manuscript stated that exo-vp-miR224 would accumulate in the recipient cells but did not present any evidence on this. The authors should either present evidence on the miR-224 accumulation or delete the statement.

**Part III – Minor Issues: Editorial and Data Presentation Modifications**

Reviewer #1: The manuscript needs to be revised to correct language problems. A list of few examples is provided below, however there are much more along the text.

Line 31 “targeting to HSP70” needs correction. In addition, in this context, “target” is a vague term, the authors should be more precise may be “inhibit” or “block” or “ downregulate the expression”.

Line 66 “are usually served as markers” needs correction

Lines 128-131 this sentence does not make much sense. May be a part of the sentence is lacking or a verb (“were isolated”??)

Line 131-132 “then to evaluate the quality of the isolated exosomes, we performed a series of tests required” not very well written needs to be improved

Line 149 “were serves as control” needs to be corrected

Line 154 “and therefore affecting the mud crab mortality” needs to be corrected

Line 212 “homeostasis of mud crab by exosomal miR-224 packaged” needs to be corrected

Reviewer #2: 5. There are many other minor errors as indicated below but not limited to:

Line 54-55: “was further interacted with” should be “further interacted with”

Line 66: remove “are” in “are usually served as”

Line 76: remove “very” from “very little is…”

Line 107: “uncontrol proliferation” should be “uncontrolled”

Line 129: “exosomes isolated from” should be “exosomes were isolated from”

Line 149: remove “were” from “were served”

Line 207: “A similarly trend” should be “A similar trend”

Line 214-215: change “the target genes controlled by miR-224” into “the genes targeted by miR-224”

Line 226: change “the both two software” into “both software”

Line 276: change “it was obviously” into “it was obvious”

Line 294-295: change “in the ubiquitination of Ecsit inhibit” into “in the inhibition of Ecsit ubiquitination"

PLOS authors have the option to publish the peer review history of their article (what does this mean?). If published, this will include your full peer review and any attached files.

Reviewer #1: No

Reviewer #2: No

Figure Files:

Data Requirements:

Reproducibility:

References:

---

## [Editor Report · Decision Letter 1]

24 Jul 2021

Dear Dr. Li,

We are pleased to inform you that your manuscript 'Exosomal miR-224 contributes to hemolymph microbiota homeostasis during bacterial infection in crustacean' has been provisionally accepted for publication in PLOS Pathogens.

The associate and senior editors have added notes in comments to authors of specific changes needed. Additional changes may be requested in a follow up email.

Best regards,

Karla J.F. Satchell, Ph.D.

Section Editor

PLOS Pathogens

Karla Satchell

Section Editor

PLOS Pathogens

Kasturi Haldar

Editor-in-Chief

PLOS Pathogens

orcid.org/0000-0001-5065-158X

Michael Malim

Editor-in-Chief

PLOS Pathogens

orcid.org/0000-0002-7699-2064

Note from Editors: The manuscript has been revised and both the reviewers and editors feel the manuscript is acceptable for PLoS Pathogens.

There are minor grammatical edits needed in the updated sections and Figure S1 has a typographical error that will require substitution of the figure with a corrected figure.

Please address these issues in a final version when requested to submit a corrected proof.

1. Abbreviations need to be used only when used more than twice in a manuscript. In all cases, the abbreviation should be in parentheses AFTER the word(s) define, not before.

2. Data is plural, so text should be "data were" as opposed to the frequently used "data was"

Please make these detailed changes along with any other changes requested by the publication team.

Line 65, comma after CD63

Lines 118-124 and through out manuscript.

Place the abbreviations in paranthesis following the word

Example: suppression of Heat shock protein 70 (HSP70) and disruption of HSP70-tumor necrosis factor receptor associated factor 6 (TRAF6).

Line 154 should be "suggested"

Line 184. Period after Fig. 2F. Start new sentence with Meanwhile,

Line 339, spell out GPLs rather than using abbreviation.

Line 343-4 spell out TNFalpha and IL-12 as Tumor necrosis factor alpha and interleukin-12 as there are mentioned only here and abbreviations not used again in manuscript.

Line 364, spell out WSSV

Line 365, 368, spell out apoptosis inducing factor and Hepatitic C Virus. No need to add AIF or HCV as abbreviations are never used again.

Line 752 & 772 & 787 & 790 & 811 & 814 & 837 & 850. Should be "data were presented"

Supplemental figure S1. There is a typo in the word Exosom in every panel. I think an "e" is missing. Please correct.

The legend should be a bit more specific, I think is missing that the labeled exosomes were injected to the mud crab and then hemocytes isolated and analyzed by flow cytometry.
---

## [Editor Report · Acceptance letter]

6 Aug 2021

Dear Dr. Li,

We are delighted to inform you that your manuscript, "Exosomal miR-224 contributes to hemolymph microbiota homeostasis during bacterial infection in crustacean," has been formally accepted for publication in PLOS Pathogens.

Best regards,

Kasturi Haldar

Editor-in-Chief

PLOS Pathogens

orcid.org/0000-0001-5065-158X

Michael Malim

Editor-in-Chief

PLOS Pathogens

orcid.org/0000-0002-7699-2064